# Point-Plane SLAM Using Supposed Planes for Indoor Environments

**DOI:** 10.3390/s19173795

**Published:** 2019-09-02

**Authors:** Xiaoyu Zhang, Wei Wang, Xianyu Qi, Ziwei Liao, Ran Wei

**Affiliations:** 1Robotics Institute, Beihang University, Beijing 100191, China; 2Beijing Evolver Robotics Technology Co., Ltd., Beijing 100192, China

**Keywords:** SLAM, RGB-D camera, factor graph, planes, plane edges, structural constraints, indoor environments

## Abstract

Simultaneous localization and mapping (SLAM) is a fundamental problem for various applications. For indoor environments, planes are predominant features that are less affected by measurement noise. In this paper, we propose a novel point-plane SLAM system using RGB-D cameras. First, we extract feature points from RGB images and planes from depth images. Then plane correspondences in the global map can be found using their contours. Considering the limited size of real planes, we exploit constraints of plane edges. In general, a plane edge is an intersecting line of two perpendicular planes. Therefore, instead of line-based constraints, we calculate and generate supposed perpendicular planes from edge lines, resulting in more plane observations and constraints to reduce estimation errors. To exploit the orthogonal structure in indoor environments, we also add structural (parallel or perpendicular) constraints of planes. Finally, we construct a factor graph using all of these features. The cost functions are minimized to estimate camera poses and global map. We test our proposed system on public RGB-D benchmarks, demonstrating its robust and accurate pose estimation results, compared with other state-of-the-art SLAM systems.

## 1. Introduction

Simultaneous localization and mapping (SLAM) develops quickly in recent years and becomes a fundamental problem for various applications including mobile robots, augmented and virtual reality. Various sensors can be used for SLAM such as laser-range finders [1,2] and cameras [3]. Laser-range finders provide accurate information about the environments but they are too expensive to be widely adopted. Cameras can also provide abundant information, which are much cheaper. With the availability of cheap RGB-D cameras [4,5], the depth of the scenes can also be measured more easily, especially for indoor environments.

Most of the existing methods for SLAM are based on a collection of points and use points to describe the scenes and estimate the camera poses. Points can be described by simple mathematical expressions and applied in both indoor and outdoor environments. But these methods encounter various problems in practice application, such as low-texture environments and changing light. Besides, the correct data association is also a challenge for point-based SLAM to obtain reliable estimation results. Direct methods are based on image intensities, which can be affected by changing light or viewing angles. Feature-based methods generally search corresponding points based on descriptors, so their results depend on the reliance of detecting and matching of feature points. The error from points measurement noise and data association will accumulate, especially in large scenes. These problems are hard to solve using only points.

For indoor environments, there are lots of other high-level features, such as lines and planes. Indoor environments are also common working scenes for mobile robots. These high-level features ensure faster and more accurate data association, which can be extracted easily using RGB-D cameras. The planes calculating from many points are more robust and accurate, because of less affection from measurement noise. Therefore using these high-level features helps to improve the performance of SLAM. In indoor environments, there are various man-made objects and structures, which have lots of parallel and perpendicular planes. Using these kinds of structural constraints can also help to achieve a long-term association for planes, resulting in smaller accumulated error. The plane is usually described as an infinite plane in mathematics. However, the real planes in working environments have limited size, contours or edges. Therefore, these features can also be exploited to add constraints for robust pose estimation.

In this paper, we propose a SLAM system using both points and planes to achieve robust and accurate estimation results. Using a RGB-D camera, we detect and match feature points in RGB images and generate point clouds from depth image to extract planes. Unlike other plane-based SLAM using infinite planes, we try to make use of plane edges. For indoor environments, a plane edge is generally an intersecting line of two perpendicular planes. In order to add the constraints of plane edges, we calculate and generate their perpendicular planes even when they are not seen. We also use contour points of planes to achieve robust data association. Then we use all of these points and planes to solve the poses of the camera and generate a map consisting of points and planes. Besides, we add parallel and perpendicular constraints for planes, which help reduce drift errors in indoor environments.

In summary, the contributions of our work are as follows:We exploit plane edge constraints by generating supposed perpendicular planes from them.We achieve robust data association for planes using their contour points.We add perpendicular and parallel constrains for planes, which reduce drift errors in indoor environments.We evaluate our proposed system on public datasets and achieve state-of-the-art performance, which also performs nearly in real time.

## 2. Related Work

Many different SLAM algorithms have been proposed in recent years and most of them formulate SLAM problem as a nonlinear least-squares problem [6]. The point-based SLAM tracks features across frames and builds a global map consisting of points. ORB-SLAM [7] tracks ORB feature points and uses reprojection error to estimate camera poses. Direct methods [8] use intensities error to track poses. Some other point-based SLAM methods can also obtain a sparse map [9], semi-dense map [10] or even dense map [11,12]. But all of these works may have difficulty in data association and work poorly in low-texture environments or large scenes.

In recent years, planes are also exploited to refine the performance of SLAM algorithm. Some earliest works [13,14] add planes into the extended Kalman filter (EKF) state vectors, which are computational cost because of the growing size of the dense covariance matrix [15]. Therefore, they are limited to some small scenes. Gostar et al. [16] also discuss the transition model of plane features. Taguchi et al. [17] present a framework for registration combining points and planes. CPA-SLAM [18] proposes a novel formulation for tracking camera motion using global planes in an expectation-maximization (EM) framework. Although they use soft labeling to reduce the effect of incorrect plane association, it can still be wrong in global optimization. Our proposal method for hard labeling works well to deal with these problems. Kaess et al. [19] introduce a minimal representation for infinite planes which is suitable for the least-squares estimation without encountering singularities. They also develop a fast dense planar SLAM algorithm [20]. EFs [21] proposes a novel method of reformulation of plane estimation and optimizing trajectories without explicit parametrization of planes. All of these works mentioned above use infinite planes of three degrees of freedom to simplify the representation of real planes. They may work poorly in some scenes, where the plane features are not plentiful. Considering the limited size of real planes, our work exploits the edges of planes to add more useful constraints, achieving more accurate and robust estimation. Besides, previous works ignore the structural constraints for planes, which are very useful for indoor environments.

In indoor environments, spatial structure is utilized to help simplify pose estimation or even make it more robust and accurate. Yang et al. [22] propose pop-up 3D plane model to generate plane landmark measurements in SLAM. Most of the indoor scenes are based on the Cartesian coordinate system, defined as Manhattan World (MW) [23]. Zhou et al. [24] utilize mean-shift to track dominant directions of MW and achieve drift-free rotation by decoupling the estimation of rotation and translation. Some other works [25,26,27] also exploit planes of MW to estimate drift-free rotation. These algorithms work well in some specific scenes, but they are also easy to fail because the MW assumption is not valid for some scenes. They give us an idea to use parallel or perpendicular constraints instead of three dominant directions, which can work in more scenes in indoor environments.

## 3. Proposed Methods

### 3.1. System Overview

In this subsection, we provide an overview of our proposed point-plane SLAM using supposed planes from edges, which is shown in Figure 1. Like other modern SLAM systems, ours can also be divided into two functional parts: (1) frond-end, the tracking part extracts and matches features for new captured frame, and estimates the camera pose by minimizing the error function constituted by the tracked features in the map; (2) back-end, the map management part estimates and optimizes landmarks in the environment. The theories for point-based SLAM are thorough enough, so we augment publicly available ORB-SLAM2 [7] RGB-D implementation to build our system, and focus on exploiting planes.

Global Map. The global map consists of a set of keyframes and detected landmarks, including both points and planes. The keyframes contain observed feature points and their descriptors, observed planes and supposed planes. For a point landmark, we store a list of observations and representative descriptor. Besides observations, a plane landmark also contains its contour points, edge lines and corresponding parallel or perpendicular planes, which will be explained in Section 3.3.

Tracking. The RGB-D camera (for example, Microsoft Kinect v2) provides RGB images and depth images. We extract ORB features [28] from RGB images and match them by descriptors. From depth images, we construct 3D point clouds and then extract planes. For indoor environments, to achieve more accurate and robust implementation, we also detect edges of the plane to calculate supposed planes and add constraints between parallel or perpendicular planes. With matched points and planes from the last frame or local map, the camera pose can be estimated.

Map Management. From keyframes, a local map consisting of plans and points is constructed and updated during the local mapping. We also process a global optimization after loop closing to construct a consistent global map. The loops are detected using the bag of words based on ORB features.

### 3.2. Preliminaries

We represent the pose of the frame *k* with respect to the world coordinate system *w* by $T_{kw}\in \mathrm{SE}\left(3\right)$, which is also a rigid transformation that transforms a 3D point $P_{w}$ from the world to the camera coordinate system:(1)$$P_{k}=T_{kw}P_{w}$$

The point in Equation (Equation 1) is represented by homogeneous coordinates $P={\left(p_{1},p_{2},p_{3},p_{4}\right)}^{⊤}\in P^{3}$, and the corresponding Euclidean point is $p={\left(p_{1}/p_{4},p_{2}/p_{4},p_{3}/p_{4}\right)}^{⊤}\in R^{3}$. When a 3D point p is observed by the camera, there is a corresponding 2D pixel $u={\left(u,v\right)}^{⊤}$ in the image. Here, *u* and *v* define the position of the pixel in the image. For aligned RGB and depth images, the same point locates at the same position. The projection of p onto the image is u=ρ(p), and the back-projection is $p=\rho^{-1}\left(u\right)$.

We parametrize planes using the Hessian form $\pi ={\left(n^{⊤},d\right)}^{⊤}$, where $n={\left(n_{x},n_{y},n_{z}\right)}^{⊤}$ is the unit vector representing the plane’s orientation and *d* is the distance of the plane from the origin [29]. A point p lies on the plane π gives:(2)$$n^{⊤}p+d=0$$

Similarly, a plane in the world frame can also be transformed into the camera frame:(3)$$\pi_{k}=T_{kw}^{-⊤}\pi_{w}.$$

### 3.3. Plane Features

We construct a global map that consists of all plane features in the scenes. Every plane is segmented from organized point clouds generated from depth images. Considering the limited size of real planes, we also exploit constraints of plane edges. We calculate a supposed plane from the plane edge, which may also be observed by other frames. We try to match the plane with all planes in the map and use plane-to-plane constraints to estimate and refine the camera poses.

#### 3.3.1. Plane Segmentation

The RGB-D camera provides RGB images and aligned depth images. In a depth image, each pixel relates to a distance between the image plane and the corresponding object in the RGB image. So we can recover the structure using the camera model to back-project the pixel and we use the pinhole camera model [29] in our work:(4)$$\left(\begin{array}{c} x\\ y\\ z \end{array}\right)=d\left(\begin{array}{ccc} f_{x}^{-1} & 0 & -c_{x}f_{x}^{-1}\\ 0 & f_{y}^{-1} & -c_{y}f_{y}^{-1}\\ 0 & 0 & 1 \end{array}\right)\left(\begin{array}{c} u\\ v\\ 1 \end{array}\right),$$
where $u={\left(u,v\right)}^{⊤}$ is the valid pixel in depth image and *d* is the value in the depth image. $p={\left(x,y,z\right)}^{⊤}$ is the corresponding 3D point. $f_{x}$ and $f_{y}$ are focal length of the camera, and $\left(c_{x},c_{y}\right)$ is the camera center coordinate.

The point clouds generated from depth images are organized, having an image like grid structure. Organized structure enables fast plane segmentation from the point cloud. We follow the work of [30], which segments point clouds from RGB-D data in near real-time. In this process, we can also obtain the contour of the segmented plane. The contour will be useful for calculating supposed planes and obtaining robust data association of planes.

#### 3.3.2. Supposed Plane Serving as Edge Constraint

As we have described, the plane can be parametrized by simple mathematical expression $\pi ={\left(n^{⊤},d\right)}^{⊤}$. π represents an infinite plane, which has three degrees of freedom. It means the plane can slide along the vertical direction of its normal. But the real planes in the scenes are not infinite and they have boundaries and edge lines. Therefore, using only these four parameters without edges loses other information from the real planes. Moreover, usually few planes can be observed in one frame, resulting in insufficient constraints for pose estimation. So we need to exploit more planes or constraints from their edges to estimate camera poses in real scenes.

In indoor environments, most man-made objects or structures have regular shapes, especially those objects that have large enough plane features. Therefore, it becomes easy to extract edge lines from these segmented planes. The edge of a plane can also be seen as an intersecting line with another plane. Besides, these two planes are generally perpendicular to each other. To add constraints from plane edges, we calculate and generate a supposed perpendicular plane from every edge line, instead of adding line-based constraints directly. For every captured frame, we not only segment planes from the depth image, but also calculate such supposed planes if there are valid plane edges. Note that, supposed planes may be also observed in other frames.

When segmenting planes from organized point clouds, we can also acquire the contour of segmented planes. We extract edge lines from contour using RANSAC [31]. If the inliers are sufficient (more than 15 percent of the contour points in our experiments), the extracted line is valid. Then we examine the position of the line to avoid the border of the image. We also examine the points near the lines to remove those lines extracted from shadow borders. A valid edge line can be represented by
(5)$$l={\left(p_{l}^{⊤},n_{l}^{⊤}\right)}^{⊤}={\left(p_{x},p_{y},p_{z},n_{x},n_{y},n_{z}\right)}^{⊤},$$
$p_{l}^{⊤}$ is a point on this line, and $n_{l}^{⊤}$ is the direction vector of the line.

The supposed perpendicular plane is calculated using the plane and its edge line. Having the representation of the plane $\pi_{i}={\left(n_{i}^{⊤},d_{i}\right)}^{⊤}$ and its edge line $l_{j}={\left(p_{j}^{⊤},n_{j}^{⊤}\right)}^{⊤}$, the supposed plane is:(6)$$\pi_{supposed}=\left(\begin{array}{c} n_{ij}\\ d_{ij} \end{array}\right)=\left(\begin{array}{c} n_{i}\times n_{j}\\ -n_{ij}^{⊤}p_{j} \end{array}\right)$$

The process of generating supposed planes is shown in Figure 2. It is a frame from the sequence ‘freiburg3_cabinet’ of Technical University of Munich (TUM) RGB-D [32] dataset. There is a cabinet in the image, and three planes of the cabinet are extracted. We know there are five planes which can be observed, except for the bottom plane. But only three planes can be observed at most in one frame, although we can imagine the other two planes. To exploit more constraints of such common man-made objects, we suppose planes (the green planes in Figure 2c) from the plane edges. We remove those repeating planes. When the camera goes to observe in the opposite direction, these supposed planes will be observed.

We can treat supposed planes as ordinary planes, like those observed by the camera directly. Plane landmarks are also created from supposed planes and may be observed by other frames. Therefore, supposed planes increase the number of planes and add constraints from plane edges. Besides, supposed planes have a perpendicular orientation to the corresponding planes. Planes of different orientations provide more sufficient constraints for accurate pose estimation.

#### 3.3.3. Data Association

Fast and accurate data association is the guarantee for robust and accurate pose estimation. Feature points can be associated with descriptors. For planes, previous work [18,19] find their correspondences just by normal $n^{⊤}$ and distance *d*. This method does work for simple environments. But it also depends on an accurate pose estimation of the camera, which is not often satisfied, especially when the current pose is predicted from the previous pose. When the angle of the plane has some noise, the distance usually fluctuates largely, leading to association failure.

We implement a novel data association algorithm for matching planes. First, we find those intersecting planes on the map. In indoor environments, intersecting planes are corresponding planes with a small-angle difference because of noise, or different planes (usually perpendicular planes) with a large-angle difference. Therefore, we just need to check the angle of these intersecting planes to find correspondence, and the angle threshold can be a little larger (30° in our experiments).

To find intersecting planes in the global map, we calculate the distances from the points on the observed planes to the plane landmarks $\pi_{i}={\left(n_{i}^{⊤},d_{i}\right)}^{⊤}$. We only examine the points $p_{j}$ from the contour $C_{m}$ acquired during plane segmentation to reduce the calculating amount:(7)$$s=\min\{|n_{i}^{⊤}p_{j}+d_{i}\left|\right\},p_{j}\in C_{m}.$$

If *s* is smaller than the distance threshold (0.1 m in our experiments), these two planes are considered intersecting planes. Then we examine the angle of these two planes to determine the corresponding plane. For every plane observation, we try to find a corresponding plane with the smallest *s* in the global map. If a plane observation fails to be associated, it is added to the global map as a new plane landmark.

Besides corresponding planes, we also exploit parallel and perpendicular planes. For every plane, we find a parallel plane if their intersection angle is small enough (10° in our experiments) but the distance *s* between them is large (larger than 0.1 m in our experiments). If there are several planes, we choose the plane having the smallest intersection angle. Similarly, we find a perpendicular plane if the angle between them is large enough (80° in our experiments).

Therefore, for every plane observation, we try to find one corresponding plane, one parallel plane and one perpendicular plane on the global map.

### 3.4. Tracking Using Points and Planes

It is well known that SLAM problem can be represented as a factor graph G(V,E) [33]. The vertices V represent the variables to estimate, such as camera poses and landmarks. The edges E between vertices represent the constraints. The factor graph enables an insightful visualization of SLAM problem. The process of solving SLAM problem is to construct such a factor graph and minimize the errors of all involved factors.

#### 3.4.1. Factor Graph Containing Points and Planes

As mentioned before, we extract ORB features in the RGB image for the current frame and find matched point landmarks using ORB descriptors. A simple point-based SLAM can be represented as a factor graph in Figure 3a. The factors are constraints of the reprojection error:(8)$$f_{z}\left(p_{w},T_{cw}\right)={∥u_{m}-\rho \left(T_{cw}P_{w}\right)∥}_{\Sigma_{z}},$$

$T_{cw}P_{w}$ is the 3D point in the camera coordinate system and $u_{m}$ is the corresponding pixel in the current frame. ${∥x∥}_{\Sigma }$ is the mahalanobis norm, which equals $x^{⊤}\Sigma^{-1}x$, and Σ is the corresponding covariance matrix.

Likewise, planes can also be added into the factor graph as landmarks. As shown in Figure 3b, *m* means the direct observation of planes and sp denotes the plane is supposed in the frame. Therefore, $\pi_{2}$ is a supposed plane in frame $T_{2}$, but it can be observed in frame $T_{3}$ and $T_{4}$.

Now we need to define the factor connecting the vertices V(T) and V(π). Notice that the Hessian form $\pi ={\left(n^{⊤},d\right)}^{⊤}$ is an over-parameterization of planes, because a 3D plane has only three degrees of freedom. Therefore, the Hessian form requires extra constraints to ensure the unit length of the plane normal vector, adding additional computation in optimization. To overcome this problem, we choose minimal parameterization of planes in optimization τ=(ϕ,ψ,d), where ϕ and ψ are the azimuth and elevation angle of the normal respectively,
(9)$$\tau =q\left(\pi \right)={\left(\phi =\arctan\frac{n_{y}}{n_{x}},\psi =\arcsin n_{z},d\right)}^{⊤},$$

The azimuth and elevation should be restricted in (−π,π] to avoid the singularities of the minimal representation in optimization. Now we can define the factor for planes:(10)$$f_{m}\left(\pi_{w},T_{cw}\right)={∥q\left(\pi_{m}\right)-q\left(T_{cw}^{-⊤}\pi_{w}\right)∥}_{\Sigma_{m}},$$
where $T_{cw}^{-⊤}\pi_{w}$ is the 3D plane in the camera coordinate system. $\pi_{m}$ is the corresponding plane observation in the current frame. Note that the covariance should be a little larger for those supposed planes, like $sp_{1}$ in Figure 3b.

Besides, we also add the structure factors E(S) for indoor environments. Structure factors, parallel or perpendicular constraints between planes, add more edges between planes, as shown in Figure 3c. The structure factors only add normal constraints of planes. For parallel constraints:(11)$$f_{sp}\left(\pi_{w},T_{cw}\right)={∥q_{n}\left(n_{m}\right)-q_{n}\left(R_{cw}n_{w}\right)∥}_{\Sigma_{sp}},$$
where $q_{n}$ means the azimuth and elevation angle in Equation (Equation 9) and n is the normal of the plane. $R_{cw}$ is the rotation of the current frame. Note that, if normal vectors point to opposite directions, we need to rotate them to the same direction first.

Similarly, the factors for perpendicular planes:(12)$$f_{so}\left(\pi_{w},T_{cw}\right)={∥q_{n}\left(R_{⊥}n_{m}\right)-q_{n}\left(R_{cw}n_{w}\right)∥}_{\Sigma_{so}}.$$

The only difference is an additional rotation matrix $R_{⊥}$ to rotate the normals to the same direction.

#### 3.4.2. Pose Estimation

For every new captured frame, we extracted feature points from the RGB image, planes and supposed planes from the depth image. If the last frame is tracked successfully, we use a constant velocity model to predict the pose of the current frame and search point landmarks observed in the last frame. Corresponding planes, parallel and perpendicular planes can also be searched in the global map. Because there are not too many planes in scenes, this search simple method is efficient enough. With matched points and planes, the current pose is then optimized. We also try to search more point correspondences in the local map. The local map contains several keyframes sharing point and plane landmarks with the current frame. The current pose is finally optimized with all points and planes in the local map.

With tracked points and planes, pose $T_{cw}$ can be computed by solving:(13)$$T_{cw}=\underset{T_{cw}}{\arg\min}\left(\sum H_{z}\left(f_{z}\right)+\sum H_{m}\left(f_{m}\right)+\sum H_{sp}\left(f_{sp}\right)+\sum H_{so}\left(f_{so}\right)\right),$$
where $H\left(x\right)$ is the Huber robust cost function and *f* are factors described in Section 3.4.1. We use the Levenberg–Marquadt method implemented in g2o [34] to solve this equation. The Jacobians of these components in Equation (Equation 13) are described detailedly in Appendix A. To remove bad observations, we check the error value of every factor after optimization and delete those larger than the threshold.

#### 3.4.3. Keyframe Decision

The keyframes are used to construct the local and global map. Besides, the loop detection and relocalization are implemented based on keyframes. We utilize the new keyframe criteria in [7]. If a successfully tracked frame observes a new plane landmark or tracks less than 90% points of the last keyframe, it will be labeled as “keyframe”. If $N_{f}$ frames have passed from the last keyframe, it will also be constructed as a keyframe. We set $N_{f}$ smaller than the output frequency of the RGB-D camera to ensure at least one keyframe will be inserted in one second. Those redundant keyframes will also be deleted later according to the number of landmarks observed.

### 3.5. Map Management

Map management is the back-end part of the whole system. It constructs a consistent global map and the camera poses are further optimized.

#### 3.5.1. Local Mapping

Every time the tracking part inserts a keyframe, new point landmarks are created by triangulation. A new plane landmark is also created if a new plane is observed or supposed. The local optimization optimizes the camera poses and landmarks on the local map. The local map contains a set of keyframes sharing landmarks with the currently processed keyframe, all of the landmarks observed by these keyframes and the parallel or perpendicular planes in the map. Other keyframes that also observe the landmarks in the local map are also included in the local optimization to provide sufficient constraints but remain fixed. The cost function is also created using all of the factors described in Section 3.4.1.

After local optimization, we delete those redundant keyframes, whose 90% of the observed points and all of the observed planes have been observed by other keyframes. The bad points and planes that have large errors are also deleted.

#### 3.5.2. Loop Closing and Gobal Optimization

We compute the bags of words representation of every keyframe to detect loops. The loop detection part is implemented using DBow2 [35]. Once a loop is detected, we perform a pose graph optimization. All the camera poses, points and planes in the global map will also be optimized in the global optimization. The global optimization uses the same factors as local optimization but has all the vertices in the global map. The drift error is reduced after the global optimization.

## 4. Experiments

We evaluate our proposed SLAM system using the benchmarks TUM RGB-D dataset [32] and ICL-NUIM deteset [36]:TUM RGB-D dataset is a famous benchmark for evaluating vSLAM/VO systems. It contains a large set of image sequences recorded from a RGB-D camera. The RGB and depth images are captured with a 640 × 480 resolution at the video frame rate (30 Hz). The ground truth camera poses are also provided. The dataset covers various indoor scenes and we choose those having obvious plane landmarks to evaluate our SLAM system.ICL-NUIM dataset is a synthetical benchmark. The images are captured within synthetically generated indoor environments, including living room and office scenes. These scenes are suitable for our point-plane SLAM system. It also provides ground truth like the TUM RGB-D dataset, which is convenient to evaluate the results.

For implementation, our system augments the RGB-D variant of ORB-SLAM2 [7]. We rely on the underlying ORB-SLAM2 for points extraction and matching. We also use the methods in ORB-SLAM2 to maintain a local map and detect loops. The focus of our implementation is on the plane extraction, supposed plane calculation, plane matching, and pose estimation using points and planes. We construct a global map consisting of points and planes, including supposed planes. All experiments run on a laptop computer with i7-7700HQ 2.80GHz CPU, 16GB RAM, without GPU.

We compare our proposed SLAM system with some other RGB-D SLAM systems. ORB-SLAM2 [7] is the state-of-the-art feature-point based visual SLAM system and it has a RGB-D implementation. L-SLAM [25] is a RGB-D SLAM system using planes and MW constraints. Note that we test ORB-SLAM2 using the open-source code provided by the author and we include the results of L-SLAM from [25] directly. To demonstrate the benefit of supposed planes, we compare the results of different versions of our proposed SLAM system, which is point-plane SLAM (PP), point-plane SLAM using structural constraints (PP+S) and point-plane SLAM adding supposed planes (PP+SS).

### 4.1. ICL-NUIM Dataset

The ICL-NUIM dataset contains several sequences from two kinds of scenes, office and living room, as shown in Figure 4. Four sequences are recorded in each scene. We run experiments on all of these sequences.

We use the root mean square error (RMSE) of the absolute trajectory error (ATE) to evaluate the performance of different SLAM systems. We report the results in Table 1. The smallest error for every sequence is labeled by the bold number. The comparison of the RMSE is also shown in Figure 5.

We first add plane-based constraints only (PP). Because there are enough planes in these indoor sequences, these additional useful constraints already help to improve the accuracy of the camera poses. When there are only a few feature points can be tracked, like sequence `living_room_1’, point-based SLAM (ORB-SLAM2) performs poorly. But the planes improve the performance of the pose estimation. The planes in these indoor scenes are usually parallel or perpendicular, therefore these structural constraints (PP+S) improve the accuracy further.

Besides, we also add supposed planes (PP+SS). Table 2 shows the average number of observed planes for every frame. In some sequences, such as ‘living_room_0’ and ‘office_room_1’, the estimated trajectories are similar to those of ‘PP+S’, because few supposed planes can be utilized. In these sequences, the plane features are generally extracted from the wall, or planes from the furniture are too small to be extracted because the camera is far from them. Therefore, only a small number of supposed planes can be exploited. For other sequences, the supposed planes generated in the tracking process are shown in Figure 6. The supposed planes are generally estimated from table edges in these scenes. Some supposed planes may not be observed directly because of their small size. Therefore, supposed planes exploit more plane features in the scenes. Besides, adding supposed planes increases the number of plane observation for every frame and add more constraints. Therefore, our proposed method using supposed planes significantly reduces the estimation error, better than PP and PP+S.

Some comparisons of the estimated trajectories of our proposed methods are shown detailedly in Figure 7. They are all close to the ground truth. But the trajectories of PP have larger drift errors. This also demonstrates that structural constraints help to reduce errors effectively. Supposed planes improve estimation accuracy further.

L-SLAM utilizes MW constraints to obtain the drift-free rotation motion of the camera. Unlike our proposed method, they extract the planes parallel with manhattan axis. The ‘office_room’ sequences are suitable and L-SLAM performs best on these sequences. But our proposed method also works well and is suitable for more scenes, because lots of real scenes can not meet the MW assumption. We also calculate the weighted average error, and the number of frames serves as the weighting factor. The average error of our method is the smallest. Therefore, our proposed SLAM system improves a lot over the state-of-the-art point-based SLAM system and achieves more robust and accurate estimations in different scenes.

Some trajectories and reconstruction results of our proposed method are shown in Figure 8. Those large planes are extracted and associated accurately. The estimated trajectories are also close to the ground truth.

### 4.2. TUM RGB-D Dataset

We select several sequences containing enough plane features from the TUM RGB-D dataset to evaluate our proposed SLAM system. Because some sequences only have a few features, the original ORB-SLAM2 fails to initialize or track camera poses. Therefore, we modify some thresholds to make ORB-SLAM2 work normally, although the performance may be somewhat worse. We also calculate RMSE to compare the estimation results of different systems, as shown in Table 3. The comparison of the RMSE is also shown in Figure 9. Table 4 shows the average number of observed planes for every frame.

For ‘fr1/desk’ and ‘fr1/xyz’, abundant point landmarks ensure the good performance of ORB-SLAM2, while our proposed methods also work well. Because only few planes satisfy the structural constraints, our methods have similar results. For the other sequences, using planes reduces the drift error. It is a challenge for point-based SLAM in low-texture scenes, such as ‘fr3/str_notex_far’ and ‘fr3/str_notex_near’. Besides, there is only one cabinet in the scene in ‘fr3/cabinet’. ORB-SLAM2 drifts significantly and even fails to track poses for the entire sequence. But planes provide sufficient constraints. For every frame, the camera can observe three planes at most. Besides, when adding supposed planes, at least one supposed plane can be utilized for every frame, on average. The supposed planes also help to add constraints for other sequences, as shown in Table 4.

As shown in Figure 10, those red planes are supposed planes generated in the tracking process. These supposed planes are associated with corresponding real planes in the scene, which increase the accuracy of pose estimation. The comparison of the estimated trajectories for some sequences is shown detailedly in Figure 11. Using planes ensures the success of tracking in the entire sequence. Structural constraints reduce the drift error. Besides, the system adding more constraints from supposed planes achieves the best estimation results.

The trajectories and reconstruction results of our proposed method are also shown in Figure 12. It is clear that structural constraints and supposed planes improve the accuracy and robustness of the SLAM system.

### 4.3. Runtime Analysis

Table 5 shows the runtime analysis of our system, running on the TUM and ICL-NUIM datasets. All the testing codes are implemented in C++. All the experiments run on a laptop computer with i7-7700HQ 2.80 GHz CPU, 16 GB RAM, without GPU and the operating system is Ubuntu 18.04.

The main difference of our system from other point-based SLAM systems is adding plane features. Therefore, we evaluate the runtime of these additional components. Segmenting planes in organized point clouds is very fast, which takes about 6.8 ms for every frame. Besides, it takes about 7 ms to extract edge lines and generate supposed planes. The runtime of other components is almost the same as point-based SLAM’s. Therefore, the additional runtime of the system is about 14 ms, compared with point-based SLAM. It takes about 38 ms to process every frame, which performs nearly in real-time.

## 5. Conclusions

In this work, we propose a novel point-plane SLAM system for indoor environments. To reduce the drift error, we add structural constraints for those parallel or perpendicular planes. Unlike other plane-based SLAM work, we exploit plane edges to add more reliable constraints. We calculate a supposed perpendicular plane according to the plane and its edge line and treat it like other plane observations. Then the camera poses can be estimated by all of these features. Our proposed algorithm is tested on public benchmarks and achieves more robust and accurate estimation results. Structural constraints can add more constraints between planes, even these planes can not be observed by the same frame. Therefore structural constraints can reduce the drift error for indoor environments. Supposed planes increase the number of plane observations in one frame and add additional sufficient constraints to solve accurate camera poses. Therefore, Our proposed SLAM algorithm is suitable to use in those indoor applications.

Currently, loop detection and relocalization are performed by using feature points. Future work will exploit planes for faster loop detection and relocalization. We will also consider adding constraints between points and planes to remove those bad landmarks.

## Figures and Tables

**Figure 1 sensors-19-03795-f001:**
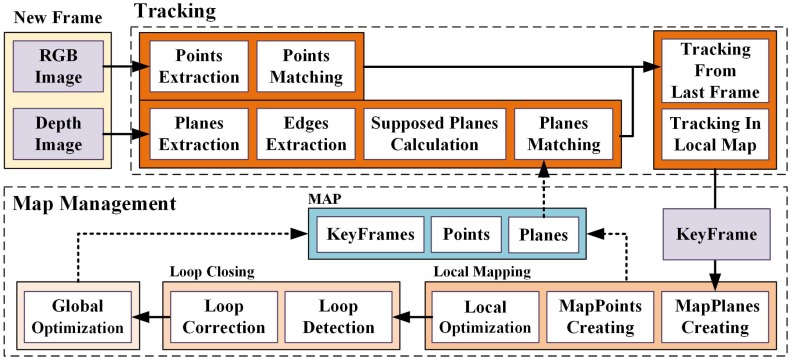
The overview of our proposed point-plane simultaneous localization and mapping (SLAM) using supposed planes from edges: the RGB-D camera provides RGB images and depth images as inputs of the whole SLAM system. The front-end tracks camera pose using matched points and planes. The back-end constructs and updates a consistent map consisting of keyframes, points and planes.

**Figure 2 sensors-19-03795-f002:**
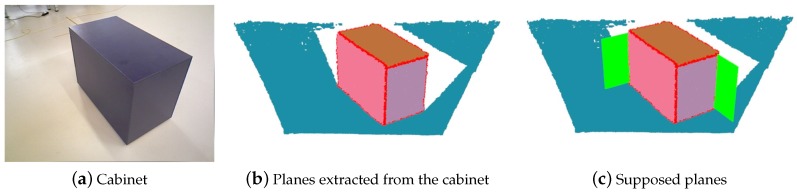
The process of generating supposed planes: (**a**) Cabinet. A frame from sequence ‘freiburg3_cabinet’ of TUM RGB-D dataset. There is only one cabinet in the image. (**b**) Planes extracted from the cabinet. Three planes of the cabinet are extracted. Different colors denote different planes. The red points are those belong to valid plane edges. (**c**) Supposed planes. The green planes denote supposed planes generated from plane edges. Other repeating supposed planes are removed.

**Figure 3 sensors-19-03795-f003:**
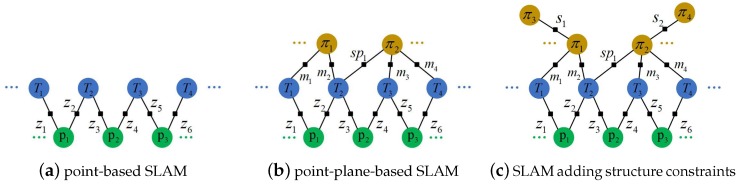
Factor graphs for SLAM. Circles denote vertices and black squares denote factors. (**a**) A simple point-based SLAM. The blue circles denote the keyframe poses and the green circles are point landmarks. (**b**) A point-plane-based SLAM. Yellow circles denote the plane landmarks. sp means supposed planes in the frame. (**c**) A point-plane-based SLAM adding structural constraints. More planes are included in the local map to add parallel or perpendicular constraints.

**Figure 4 sensors-19-03795-f004:**
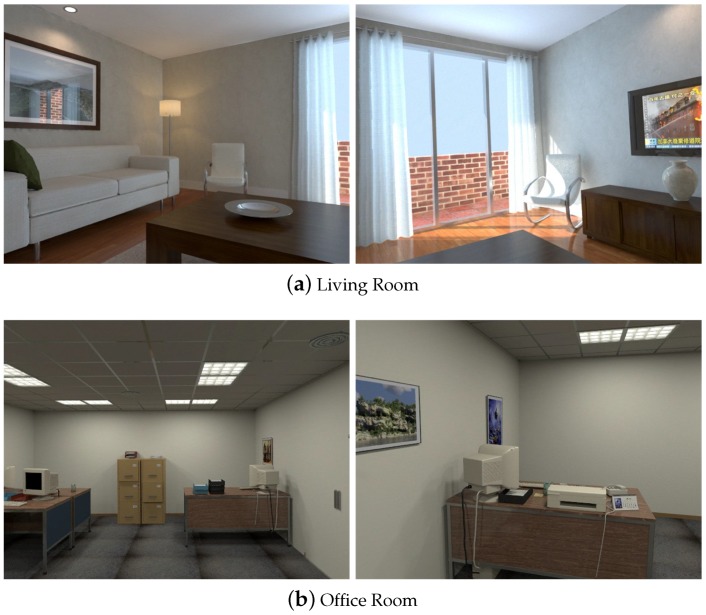
Scenes contained in the ICL-NUIM dataset. (**a**) The living room contains sofa, chairs table and some other common man-made objects. (**b**) The office room contains several tables, computers and pictures.

**Figure 5 sensors-19-03795-f005:**
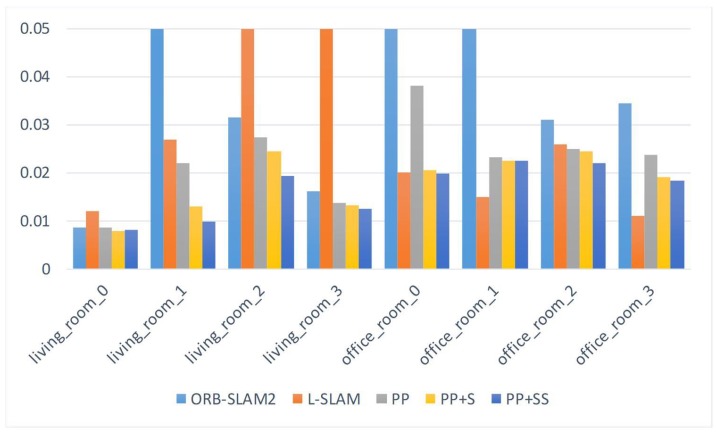
The comparison of the root RMSE on ICL-NUIM dataset.

**Figure 6 sensors-19-03795-f006:**
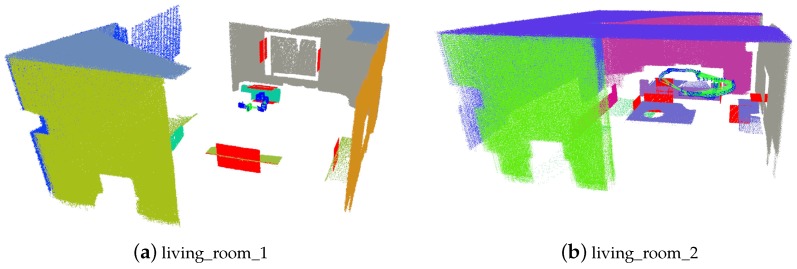
Supposed planes generated in the tracking process. Red planes denote supposed planes.

**Figure 7 sensors-19-03795-f007:**
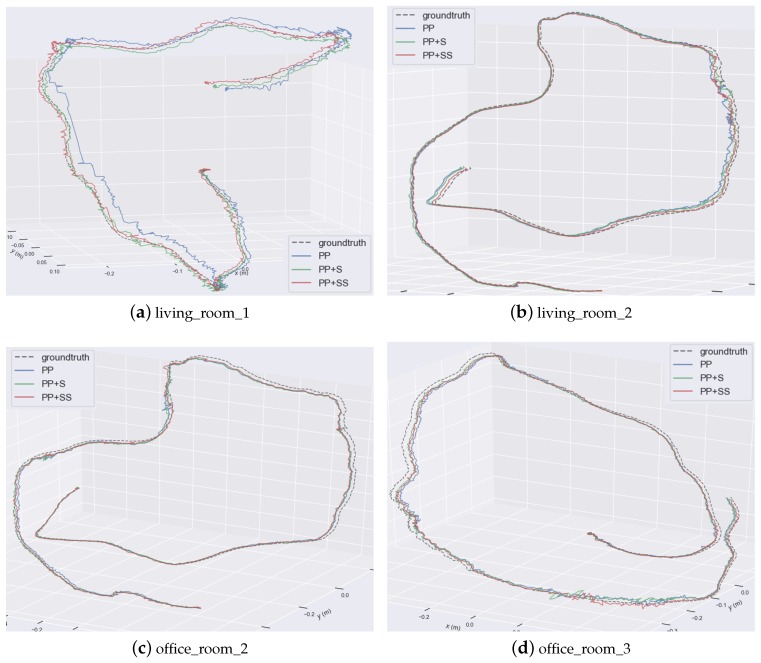
Comparison of estimated trajectories. Different trajectories are plotted together.

**Figure 8 sensors-19-03795-f008:**
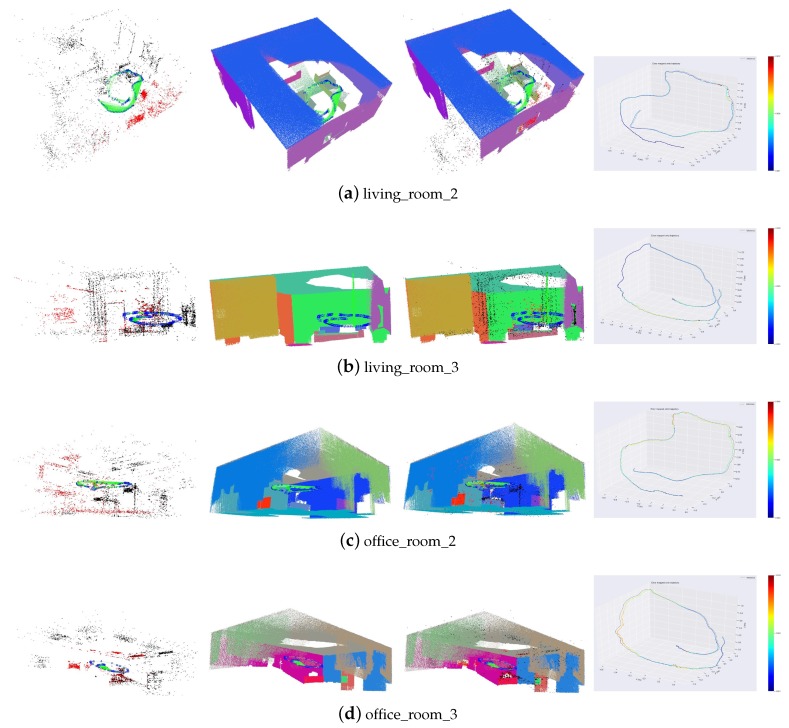
Trajectories and reconstruction results. The first column is the point landmarks and the camera trajectories. The second column shows the plane landmarks. The third column shows the plane and point landmarks together. The fourth column is the comparison of the estimated trajectories and corresponding ground truth.

**Figure 9 sensors-19-03795-f009:**
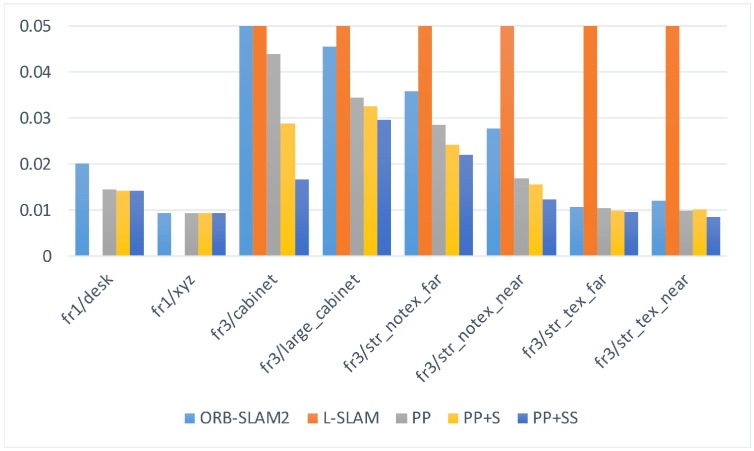
The comparison of the RMSE on TUM RGB-D dataset.

**Figure 10 sensors-19-03795-f010:**
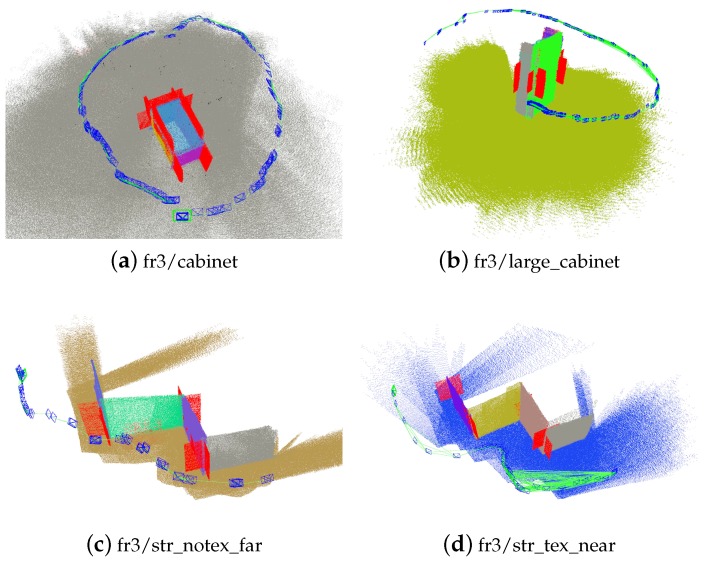
Supposed planes generated in the tracking process. Red planes denote supposed planes.

**Figure 11 sensors-19-03795-f011:**
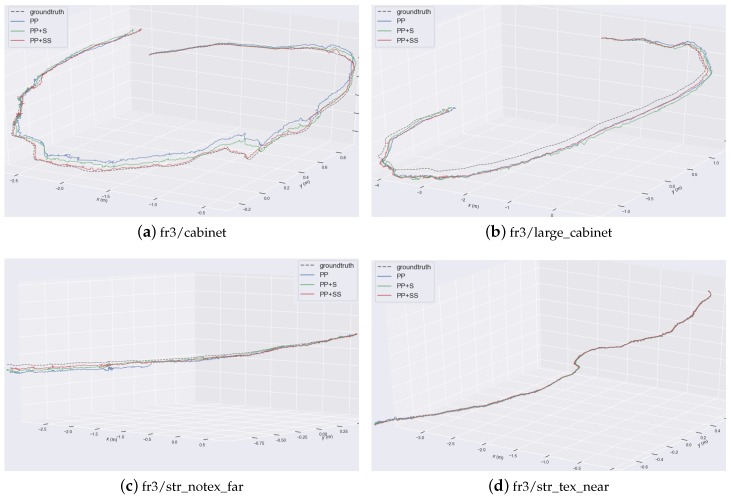
Comparison of estimated trajectories. Different trajectories are plotted together.

**Figure 12 sensors-19-03795-f012:**
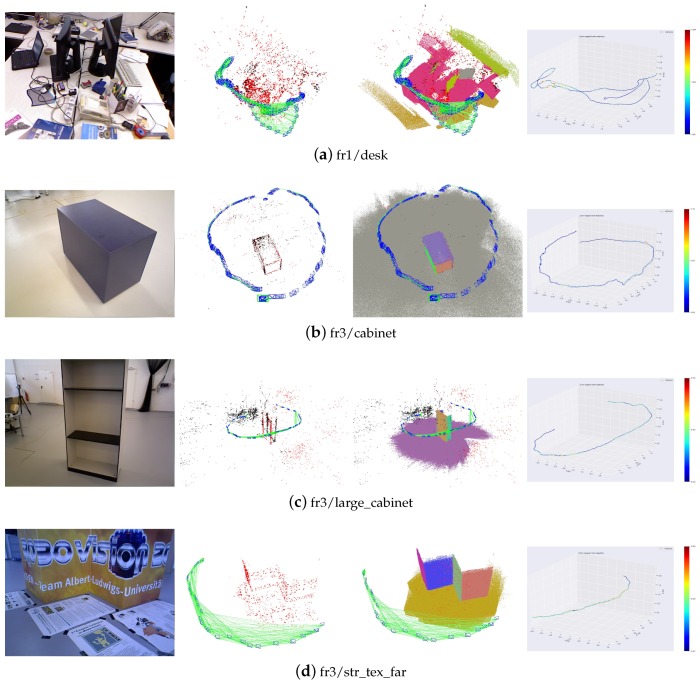
Trajectories and reconstruction results. The first column is the image showing the scene. The second column shows the point landmarks. The third column shows the plane and point landmarks together. The fourth column is the comparison of the estimated trajectories and corresponding ground truth.

**Table 1 sensors-19-03795-t001:** Evaluation results of translation absolute trajectory error (ATE) root mean squared error (RMSE) (unit: m) on ICL-NUIM dataset. PP, PP+S, PP+SS denote proposed point-plane SLAM, point-plane SLAM using structural constraints, point-plane SLAM using structural constraints and supposed planes, respectively. Bold numbers represent the best performances.

Sequence	ORB-SLAM2	L-SLAM	PP	PP+S	PP+SS	Frames
living_room_0	0.008578	0.012	0.008680	**0.007982**	0.008023	1509
living_room_1	0.200787	0.027	0.022113	0.013122	**0.009793**	966
living_room_2	0.031458	0.053	0.027399	0.024423	**0.019249**	881
living_room_3	0.016210	0.143	0.013832	0.013367	**0.012473**	1241
office_room_0	0.063775	0.02	0.038052	0.021606	**0.019861**	1508
office_room_1	0.084720	**0.015**	0.023280	0.022578	0.022546	966
office_room_2	0.030912	0.026	0.025042	0.024472	**0.022009**	881
office_room_3	0.034402	**0.011**	0.023617	0.019084	0.018483	1241
average	0.054680	0.038023	0.022517	0.017508	**0.016106**	

**Table 2 sensors-19-03795-t002:** The number of plane landmarks and the average number of observed planes on ICL-NUIM dataset. $N_{l}$ denotes the number of plane landmarks. $N_{p}$ denotes the average number of observed planes for every frame. The parameter $N_{\mathrm{sp}}$ denotes the average number of supposed planes for every frame.

Sequence	$N_{l}$	$N_{p}$	$N_{\mathrm{sp}}$
living_room_0	17	2.67197	0.082152
living_room_1	19	2.91511	0.362319
living_room_2	18	3.79115	0.589103
living_room_3	21	2.70991	0.321515
office_room_0	15	2.78515	0.043103
office_room_1	12	2.69151	0.073520
office_room_2	14	2.87855	0.178104
office_room_3	11	2.86301	0.118082

**Table 3 sensors-19-03795-t003:** Evaluation results of translation ATE RMSE (unit: m) on TUM RGB-D dataset. PP, PP+S, PP+SS denote proposed point-plane SLAM, point-plane SLAM using structural constraints, point-plane SLAM using structural constraints and supposed planes, respectively. Bold numbers represent the best performances.

Sequence	ORB-SLAM2	L-SLAM	PP	PP+S	PP+SS	Frames
fr1/desk	0.020163	-	0.014597	**0.014172**	0.014341	573
fr1/xyz	0.009377	-	0.009433	0.009530	**0.009326**	792
fr3/cabinet	0.075824	0.291	0.043939	0.028968	**0.016801**	1112
fr3/large_cabinet	0.045575	0.14	0.034616	0.032573	**0.029702**	984
fr3/str_notex_far	0.035837	0.141	0.028480	0.024204	**0.022037**	794
fr3/str_notex_near	0.027648	0.066	0.017045	0.015473	**0.012481**	1054
fr3/str_tex_far	0.010693	0.212	0.010438	0.009843	**0.009716**	907
fr3/str_tex_near	0.012139	0.156	0.010039	0.010073	**0.008481**	1057
average	0.031386	0.16927	0.021929	0.018547	**0.015391**	

**Table 4 sensors-19-03795-t004:** The number of plane landmarks and the average number of observed planes on TUM RGB-D dataset. $N_{l}$ denotes the number of plane landmarks. $N_{p}$ denotes the average number of observed planes for every frame. The parameter $N_{\mathrm{sp}}$ denotes the average number of supposed planes for every frame.

Sequence	$N_{l}$	$N_{p}$	$N_{\mathrm{sp}}$
fr1/desk	7	1.26876	0
fr1/xyz	4	1.45707	0
fr3/cabinet	6	3.06745	1.281470
fr3/large_cabinet	6	2.04268	0.357724
fr3/str_notex_far	6	2.85264	0.618388
fr3/str_notex_near	5	1.72486	0.299810
fr3/str_tex_far	6	2.94377	0.502756
fr3/str_tex_near	6	2.22422	0.284768

**Table 5 sensors-19-03795-t005:** Average runtime (unit: ms) of different components.

Main Components	Runtime (ms)
ORB Extraction	11.54862
Plane Segmentation	6.803255
Supposed Plane Generation	7.060935
Matching and Tracking Landmarks	11.87778
Local Optimization	141.1678
Global Optimization (128 KFs)	344.3516
Frame Tracking	38.07953

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
