# Peer review of "Point-Plane SLAM Using Supposed Planes for Indoor Environments"

_sensors, 2019, doi:10.3390/s19173795_

Round 1

Reviewer 1 Report

The paper, entitled "Point-Plane SLAM Using Supposed Planes for Indoor Environments" proposes a point-plane slam algorithm that takes into account points, planes, their edges by the so called supposed planes and structural relation between them in typical indoor environments.

The contributions are on the supposed planes, their DA by matching contours and plane structure that reduces drift errors.
Other works already make use of structural constraints exploiting Manhattan Worlds, but not using planes. The authors acknowledge this on their related works.

In addition, the authors propose to hallucinate planes (supposed planes) to keep track of edges. This is an interesting point, since instead of optimizing over line features, which would result in independent landmark w.r.t the plane, the supposed planes correctly capture the nature of edges (lines) while maintaining the relation with the observed plane. Still, the results show that edges are not always useful and increase performance, which most likely is dependent on the number of edges observed. Maybe the authors could comment a little more on this, since the sequences are selected by having enough planes, but not quantifying this number of planes or edges used.

The structural constraints, being either parallel or perpendicular in practice gives good results for drift error, but this creates a problem on environments when these constraints do not apply strictly, creating a strong and biased prior of how the scene should look like. Is there any result on a particular sequence where this statement did not hold?

For the implementation, the authors used ORB-SLAM2, an inherit most of their processing modules and algorithm scheme. In this regard, the authors distinguish between the original orb-slam and their contributions and differences in implementation, but at some point they should clarify this better.

The results use a set of selected sequences from two well-known data-sets that support the paper claims. The tables and graphics provided give an accurate idea of what the algorithm is doing. The authors also comment on those sequences that were problematic and potential limitations of the proposed algorithm.

The authors may be interesting on alternative approaches using to planes, such as the following,

G. Ferrer, "Eigen-Factors: Plane Estimation for Multi-Frame and Time-Continuous Point Cloud Alignment". IROS, in press 2019
(http://sites.skoltech.ru/app/data/uploads/sites/50/2019/07/ferrer2019planes.pdf)

a non-parametric approach to process planar information by aligning all points observed from different views belonging to the same plane to optimize trajectories.

Some minor grammar details, I might have missed others
l24 "using which"
l27 practice use
l145 "especially those have large enough planes."
l186 angel
l214 planes are also be searched

Reviewer 2 Report

The trajectory image like figure 11 is not clear enough. The author said in page 2“which are computational cost because of the dense covariance matrix.“, please give some explanation and related literature.

3、The part of related work should be improved, the comparison with other plane-based should be emphasized.

4. The author said in the introduction“which also performs nearly in real time.“, if too many planes can be detected in the environment, the processing time of Frame Tracking will be too long and the system can’t run in real time.

5、The number of plane used for pose estimate can be counted to explain The benefit of the plane features for pose estimate clearly.6、If the jacobian of component 、 in equation 13 shown in the Appendix, the paper will be more complete.

Round 2

Reviewer 2 Report

I have no further comments about this manuscript and all my questions are properly solved.